# An Efficient and Accurate Convolution-Based Similarity Measure for Uncertain Trajectories

Guanyao Li [1,2,3], Xingdong Deng [2,3,*], Jianmin Xu [1], Yang Liu [2,3], Ji Zhang [2,3], Simin Xiong [2,3] and Feng Gao [2,3]

[1] School of Civil Engineering and Transportation, South China University of Technology, Guangzhou 510641, China; gyli@gzpi.com.cn (G.L.);aujmxu@scut.edu.cn (J.X.)

[2] Guangzhou Urban Planning & Design Survey Research Institute, Guangzhou 510060, China; liuyang@gzpi.com.cn (Y.L.); j.zhang@gzpi.com.cn (J.Z.); xiongsimin@gzpi.com.cn (S.X.); 2111801048@e.gzhu.edu.cn (F.G.)

[3] Guangdong Enterprise Key Laboratory for Urban Sensing, Monitoring and Early Warning, Guangzhou 510060, China

[*] Correspondence: dengxingdong@gzpi.com.cn

**Abstract:** With the rapid development of localization techniques and the prevalence of mobile devices, massive amounts of trajectory data have been generated, playing essential roles in areas of user analytics, smart transportation, and public safety. Measuring trajectory similarity is one of the fundamental tasks in trajectory analytics. Although considerable research has been conducted on trajectory similarity, the majority of existing approaches measure the similarity between two trajectories by calculating the distance between aligned locations, leading to challenges related to uncertain trajectories (e.g., low and heterogeneous data sampling rates, as well as location noise). To address these challenges, we propose Contra, a convolution-based similarity measure designed specifically for uncertain trajectories. The main focus of Contra is to identify the similarity of trajectory shapes while disregarding the time/order relevance of each record within the trajectory. To this end, it leverages a series of convolution and pooling operations to extract high-level geo-information from trajectories, and subsequently compares their similarities based on these extracted features. Moreover, we introduce efficient trajectory index strategies to enhance the computational efficiency of our proposed measure. We conduct comprehensive experiments on two trajectory datasets to evaluate the performance of our proposed approach. The experiments on both datasets show the effectiveness and efficiency of our approach. Specifically, the mean rank of Contra is 3 times better than the state-of-the-art approaches, and the precision of Contra surpasses baseline approaches by 20–40%.

**Keywords:** trajectory similarity; uncertain trajectory; location noise; low data sampling; heterogeneous data sampling

## 1. Introduction

With the prevalence of mobile devices and the rapid development of localization techniques, object location could be extracted based on various ubiquitous signals, including GPS, Wi-Fi, Bluetooth, video, etc. [1–3]. Furthermore, location data are collected when individuals utilize different services, such as mobile payment methods (e.g., credit cards, Apple Pay, WeChat Pay, etc.), navigation apps (e.g., Google Maps, Baidu Maps, AutoNavi, etc.), and online-to-offline services (Didi, Dianping, Foursquare, etc.) [4]. Consequently, trajectory data, which represent sequential records of locations, are being generated at an unprecedented pace.

Due to its immense commercial and social significance, trajectory analytics has become increasingly valuable in a wide range of domains, including user analytics, pandemic prevention and control, urban planning, transportation, etc. [5–8]. At the core of trajectory analytics lies the concept of trajectory similarity, which quantifies the spatial overlap between trajectories.

In the field of user analytics, trajectory similarity can be used to identify similar movement patterns among users, which is valuable for personalized recommendations, targeted marketing campaigns, and understanding user behavior [9]. Another crucial application is pandemic prevention and control. Trajectory similarity can aid in contact tracing efforts, identifying individuals who may have come into close proximity with an infected person. By analyzing the trajectories of individuals, health authorities can identify high-risk areas, track potential transmission routes, and implement targeted interventions [10,11]. Moreover, trajectory similarity plays a crucial role in urban planning by analyzing the movement patterns of individuals within a city. By comparing trajectories, urban planners can identify popular routes, understand commuting patterns, and optimize transportation infrastructure accordingly. This information helps in designing efficient public transportation systems, identifying areas for new infrastructure development, and improving overall urban mobility [12–14].

An uncertain trajectory refers to a trajectory with uncertainties, specifically location noise, low data sampling rate, and heterogeneous rate. Comparing the similarity between uncertain trajectories is significantly challenging, as it involves addressing the following issues of spatial and temporal uncertainties:

- Heterogeneous sampling rates: Due to the nature of sensing devices and object activities, trajectories are sampled from continuous paths of objects with time-varying heterogeneous sampling rates, i.e., locations in a trajectory are collected randomly and sporadically. As a result, trajectories sampled from the same path may not share overlapping locations, making it challenging to accurately measure their similarity.
- Low sampling rates: Some trajectories could be very sparse due to the low data sampling rate of the sensing system (e.g., call detail records in a telecommunication system). The time interval between two consecutive locations could be large (such as tens of minutes), and object locations are not observed during that time. The infrequent observations of object locations introduce uncertainty when measuring trajectory similarity.
- Location noise: The location in a trajectory is not always perfectly accurate because of the device error or other environmental factors [15,16]. Consequently, objects that are actually co-located may appear at different positions, affecting the measurement of similarity.

Considerable research has been dedicated to trajectory similarity, as evident in references [17–21]. Most existing methods measure similarity by aligning locations from two trajectories and utilizing the distance between these locations as a measure of trajectory similarity. Although effective in certain scenarios, their performance declines when encountering challenges associated with trajectory uncertainty. Furthermore, some techniques rely heavily on extensive data for model training [22,23], which may not always be available.

To address the limitation of existing works, we propose Contra, a convolution-based similarity measure specifically designed for uncertain trajectories with location noise, as well as low and data sampling rates. Note that Contra focuses on identifying the similarity of trajectory shapes while disregarding the time/order relevance of each record within the trajectory. Unlike the conventional approaches that align locations in two uncertain trajectories, Contra takes a different approach. It extracts high-level spatial features from trajectories and evaluates their similarity based on these extracted features, thereby offering a unique perspective.

The framework of Contra is illustrated in Figure 1. It begins by dividing the area into non-overlapping grids and transforming a trajectory into a matrix using a location mapping module. Subsequently, it applies a series of trajectory convolution and pooling operations using carefully designed trajectory kernels to extract high-level features. Trajectory similarity is then measured based on these extracted features. To accelerate the feature extraction and facilitate similarity comparison, we propose an efficient trajectory index for the trajectory matrix. This trajectory index enables linear-time processing of

similarity comparison. Moreover, the convolution operation for trajectory representation can be conducted offline, making Contra even more efficient.

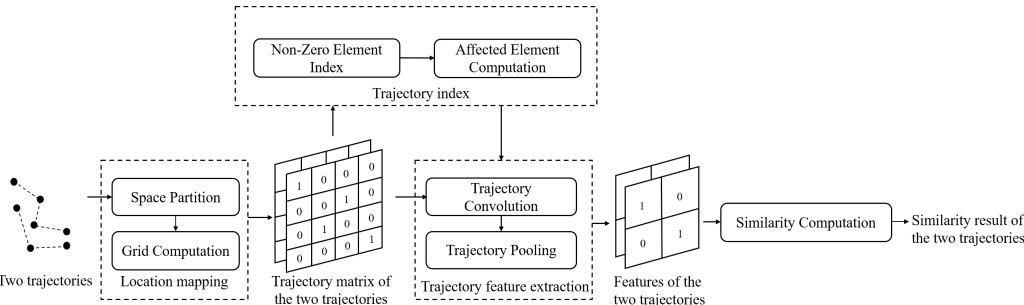

**Figure 1.** Overview of Contra.

In particular, to tackle the challenges of trajectory uncertainties, Contra compares trajectory similarity based on their high-level shape features, rather than relying on point-wise distance calculations between individual locations in the trajectories (such as prior works [18,20,21,24,25]). To this end, it exploits trajectory convolution and trajectory pooling to extract high-level features from a trajectory. The trajectory convolution technique considers the spatial dependencies among neighboring grids by aggregating features from these grids to capture high-level shape features. By doing so, this approach remains robust even when a location is incorrectly mapped to a grid due to noise. Moreover, the extracted high-level shape features could mitigate the effect of low data sampling rates. In terms of heterogeneous data sampling rates, the trajectory pooling operation enables the extraction of key features, hence mitigating the effect of data sampling rate imbalances.

Furthermore, based on the observation that a trajectory matrix could be sparse, we devised an efficient trajectory index to accelerate trajectory feature extraction and similarity computations. Compared with most existing approaches with a computation complexity of $O(M \times N)$, where $M$ and $N$ represent the number of locations in the two trajectories, our proposed approach achieves linear-time processing with the assistance of the trajectory index strategy. Linear processing time is particularly advantageous when performing tasks such as clustering, classification, or retrieval of similar trajectories. These operations often involve comparing a target trajectory against a large number of reference trajectories.

We conduct comprehensive experiments on two real-world trajectory datasets to evaluate the performance of Contra in comparison to state-of-the-art approaches. The evaluation results from both datasets demonstrate that Contra surpasses the baseline methods in terms of efficiency, robustness, and scalability for trajectory similarity measurement. The mean rank of Contra demonstrates a remarkable improvement compared to state-of-the-art approaches by up to three times, and its precision outperforms baseline approaches by an impressive margin of 20% to 40%.

The remainder of this paper is organized as follows. Section 2 discusses related works. Section 4 introduces the details of our proposed approach for trajectory similarity measurements. The experimental settings and results are shown in Section 5. Section 6 concludes this paper.

## 2. Related Work

Measuring the similarity between two trajectories has attracted much attention since it is the foundation for many research problems in trajectory mining [5]. Some works [26,27] assume that the trajectory lengths are equal. However, this assumption renders these metrics inapplicable in scenarios where there is heterogeneity in data sampling rates and varying trajectory lengths. Most existing works measure trajectory similarity by computing the distance between the aligned locations from trajectories, such as DTW [18], Fréchet distance [24], EDR [21], LCSS [20], etc. They align a location in a trajectory to a corresponding location in another trajectory using dynamic programming algorithms, and

the similarity between two trajectories is then measured by the distance between these aligned locations. However, the quadratic complexity of these approaches limits their application for large-scale trajectory similarity computation. Moreover, these approaches encounter challenges when dealing with location noise and variations in data sampling rates, making it difficult to achieve accurate alignments.

To mitigate the effects of low and heterogeneous data sampling rates, some works propose trajectory complement approaches. EDwP [19] and STED [28] employ linear interpolation for trajectory complement, based on a strong assumption that objects do not change their direction or speed between any consecutive locations in the trajectory. In addition, CATS [25] introduces temporal and spatial parameters for location-pair mapping. However, its performance heavily relies on manual parameter setting. Furthermore, APM [29] and some other works (such as [30,31]) use historical data to learn the transition probability between locations, and utilize the Markov model to estimate object location for trajectory complement. STS [32] employs a kernel density estimation method to estimate the probability distribution of object locations. However, the computational complexity of STS is high due to the need to compute a personalized transition model for each trajectory. In comparison, our approach does not rely on any prior assumptions about the trajectories and achieves efficiency through the proposed trajectory index strategies.

With the advancements of deep learning techniques in recent years, deep representation learning methods [22,23,33–36] have been successfully applied to address the issues of trajectory uncertainty for measuring trajectory similarity. In particular, bidirectional long short-term memory (Bi-LSTM) [35] and one-dimensional convolutional neural network (1D-CNN) with a long short-term memory (LSTM) network [36] were employed in prior works to model trajectory uncertainty. Moreover, t2vec [22] proposes a sequence-to-sequence framework, while GTS [23] leverages the graph neural network and LSTM to learn trajectory embedding and compare the similarity based on the learned embedding. Furthermore, to capture the long-term dependencies for similarity computation, a novel graph-based method called TrajGAT was proposed in a recent work [33]. However, these methods ignore the point-level differences between trajectories and only capture the trajectory-level features. To address this issue, CSTRM [34] proposes a novel contrastive model to learn trajectory representations for similarity computation. It captures both trajectory-level and point-level features while maintaining robustness to non-uniform sampling rates and data noise. While these works demonstrate impressive results, they heavily rely on extensive training data, which may not be available in certain scenarios. Moreover, the model must be retrained when the data distribution changes. In contrast to existing approaches, our method is training-free and does not rely on historical data, thereby expanding its applicability to a wider range of scenarios.

## 3. Preliminary

In this section, we first define path and trajectory in Section 3.1, and then overview our proposed Contra in Section 3.2.

### 3.1. Path and Trajectory

**Definition 1.** *(Path) A path refers to the actual underlying route of a moving object over time. It could be described as a continuous function $f(t) = \ell$, where $t$ is a time stamp and $\ell$ is the object location at $t$.*

A trajectory can be seen as a discrete representation of the movement of an object, obtained through a sampling process from the object path:

**Definition 2.** *(Trajectory) A trajectory is represented as a sequence of locations with the timestamp $Tra = \{(\ell_1, t_1), (\ell_2, t_2), \cdots, (\ell_i, t_i), \cdots, (\ell_n, t_n)\}$, where $\ell_i$ is the object location collected at time $t_i$.*

In contrast to a path, trajectories exhibit varying sampling rates and can be subject to noise during the sampling process. Consequently, different trajectories may arise from the same underlying path, reflecting the temporal irregularities and potential inaccuracies inherent in the trajectory sampling.

### 3.2. Overview of Contra

We overview the proposed Contra in Figure 1. Contra consists of four components, which are explained as follows.

- Location mapping: Given two trajectories, Contra first partitions the space into non-overlapping grids. Then, it computes the grid position for each location to map the locations into grids, generating two trajectory matrices for the two trajectories.
- Trajectory feature extraction: Based on the two trajectory matrices, Contra uses trajectory convolution and pooling with a well-defined trajectory kernel to extract high-level shape features for the two trajectories. This process generates two feature matrices that capture crucial characteristics and spatial correlations presented in the trajectories.
- Similarity comparison: Contra compares the similarity between the two trajectories based on their feature matrices. It computes the average difference between the two feature matrices to represent their similarity.
- Trajectory index: To accelerate the feature extraction process, we developed innovative trajectory index strategies for Contra. These strategies involve indexing the non-zero elements within a trajectory matrix and determining the positions of affected elements for each non-zero element. By utilizing the trajectory index, only the affected elements need to be updated during trajectory convolution and pooling operations, resulting in improved efficiency.

## 4. Methodology

In this section, we present the details of different components of our proposed approach. We first introduce the concept of location mapping in Section 4.1. Then, we present the trajectory convolution and pooling operations that enable the extraction of high-level features in Section 4.2, followed by the description of similarity comparison in Section 4.3. Finally, we introduce the trajectory index in Section 4.4.

### 4.1. Location Mapping

By specifying the grid size as $s$ meters, and considering a rectangular space with size $(p \times s) \times (q \times s)$, we divide the space into $p \times q$ non-overlapping grids. These grids are arranged in a matrix format, with $p$ rows and $q$ columns. This approach allows us to represent the spatial area as a space matrix $S_{p \times q}$, where each element in the matrix represents a specific grid within the space. To ensure simplicity and consistency, we assume a rectangular shape for the space, with a size of $(p \times s) \times (q \times s)$. In cases where the space has an irregular shape or its size does not precisely match $(p \times s) \times (q \times s)$, we utilize padding to convert irregular shapes into rectangular spaces with dimensions of $(p \times s) \times (q \times s)$. When the space is already rectangular, no padding is necessary for location mapping. Thus, the padded space has no effect on the similarity computation.

Next, we proceed to map each location $(x_i, y_i)$ in the trajectory onto the corresponding grid where it is located. This projection ensures that each location is associated with its respective grid in the spatial matrix representation. Consequently, the trajectory itself is transformed into a matrix format.

Given a trajectory $Tra = \{(\ell_1, t_1), (\ell_2, t_2), \ldots (\ell_n, t_n)\}$ and a space matrix $S_{p \times q}$, the trajectory matrix of the given trajectory is defined as a $p \times q$ matrix, the elements of which are defined as

$$M_{i,j} = \begin{cases} 1, \text{if } \exists\, 0 < k < n, (x_k, y_k) \text{ locates in the } S_{i,j}. \\ 0, \text{ otherwise}. \end{cases} \tag{1}$$

where $S_{i,j}$ denotes the grid at the $i$-th row and $j$-th column in the space matrix.

The elements in the trajectory matrix are represented by binary values rather than the count of locations within a grid. The reason is that when comparing the similarity of two trajectories with heterogeneous data sampling rates, a trajectory with a higher data sampling rate is likely to have more locations in a grid. To alleviate the effect of heterogeneous data sampling rates and to ensure a more equitable comparison of trajectory similarity, we employ a binary value to indicate whether an object appears in a particular region in the trajectory matrix. This binary representation simplifies the trajectory matrix by focusing on the presence or absence of an object within a specific grid region.

Using a matrix to represent an uncertain trajectory for similarity comparison offers two advantages over a location sequence representation. Firstly, the grid-based approach is highly tolerant to noise. Even if a location contains noise or inaccuracies, it can still be accurately projected into the corresponding grid. In cases where a location is mistakenly assigned to a nearby grid, the trajectory convolution operation (discussed in Section 4.2) will effectively alleviate the impact of location noise. Moreover, the matrix representation preserves the spatial dependence of the locations within a trajectory. This characteristic proves valuable in addressing challenges related to low and heterogeneous sampling rates. By leveraging the spatial dependence information captured in the trajectory matrix, it becomes possible to infer objects' co-occurrences. This inference capability is particularly useful when dealing with trajectories that have sparse or irregularly sampled data.

### 4.2. Trajectory Feature Extraction

Based on the trajectory matrix, we extract high-level features for an uncertain trajectory using the trajectory convolution and pooling operations. We first define the trajectory kernel as follows.

**Definition 3.** *(Trajectory kernel) A trajectory kernel $\mathcal{K}_{k \times k}$ is a $k \times k$ matrix with all elements as 1.*

It is important to note that Contra differs from the canonical convolution neural networks (CNNs) in that it is a training-free approach. In canonical CNN, a kernel's values are learnable parameters, which significantly rely on a huge amount of data for model training. However, we define the trajectory kernel as a matrix with all elements set to 1. This design allows the kernel to indicate the presence or absence of an object within a specific grid without relying on model training.

#### 4.2.1. Trajectory Convolution

During the trajectory convolution operation, a trajectory kernel traverses the trajectory matrix from left to right and top to bottom. Every time the kernel is hovering on the trajectory matrix, a matrix multiplication will be performed between the kernel and the hovered portion of the trajectory matrix. The resulting matrix from each multiplication is stored in a new matrix, which is then passed to the subsequent phase of the operation. Given a trajectory matrix $M$ with size $p \times q$ and a trajectory kernel $\mathcal{K}_{k \times k}$, the size of the new trajectory matrix $M^{'}$ after a convolution operation will be $(p - k + 1) \times (q - k + 1)$, and the element in $M^{'}$ is calculated as

$$M^{'}_{i,j} = \sum_{u=i}^{u=i+k-1} \sum_{v=i}^{v=i+k-1} M_{u,v}.$$

(2)

The trajectory operation process is illustrated in Figure 2. As depicted, a trajectory kernel initiates from the top-left corner of the trajectory matrix and moves horizontally to the right until it reaches the right side of the matrix. It then hops down to the left side of the next row and repeats the process until the entire trajectory matrix has been traversed. Each element in the new trajectory matrix is the summation of $k \times k$ elements in the trajectory matrix from the last phase. Consequently, every element in the new trajectory matrix

represents a latent region, and its value indicates the likelihood of an object being present in that specific region.

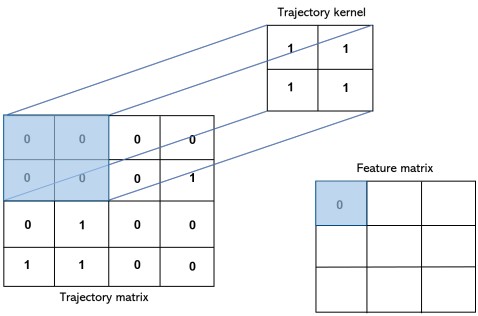

(**a**) A kernel (size = 2) starts from the top-left of the trajectory matrix.

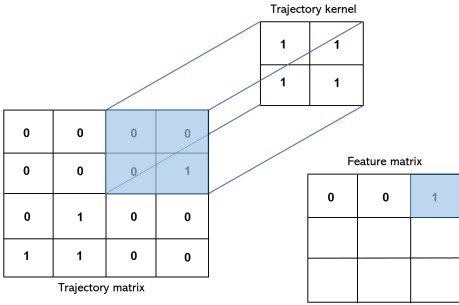

(**b**) The kernel moves to the right end of the trajectory matrix.

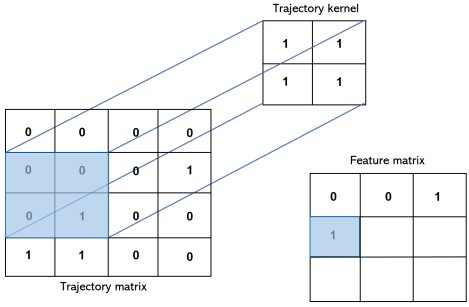

(**c**) The kernel hops down to the left of the trajectory matrix.

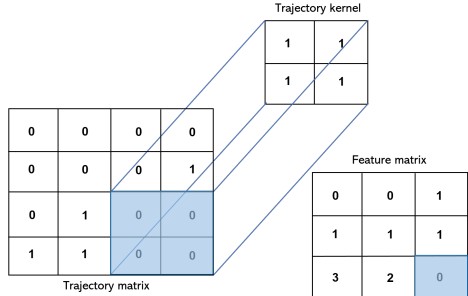

(**d**) The kernel repeats the process until the entire trajectory matrix is traversed.

**Figure 2.** Illustration of the convolution operation on the trajectory matrix.

Due to the low and heterogeneous sampling rates, it is less likely for co-located objects to be located at the exact same locations. Fortunately, the high-level features extracted through trajectory convolution contain latent region information, and similar trajectories tend to exhibit co-occurrence in certain latent regions (i.e., elements in a new trajectory matrix), even when the data sampling rates are low and heterogeneous. Moreover, the trajectory convolution operation helps alleviate the impact of location noise. When a location contains noise, it may be incorrectly projected into a nearby grid, which is in close proximity to the correct one. As the trajectory kernel considers multiple nearby grids during the convolution process, the location can still be represented in several correct latent regions. This reduces the effect of location noise as the convolution operation takes into account the spatial dependence among neighboring grids.

### 4.2.2. Trajectory Pooling

During the convolution operation, the elements in a trajectory matrix are summed if they intersect with the trajectory kernel. However, a trajectory with a higher data sampling rate will have more non-zero elements in its trajectory matrix compared to trajectories with lower data sampling rates. As a result, the values of elements in the new trajectory matrix after the trajectory convolution will be larger for trajectories with higher data sampling rates. This imbalance will potentially lead to incorrect measurements.

To address this issue, we draw inspiration from the pooling operation in CNNs, and introduce the concept of trajectory pooling. Similar to trajectory convolution, trajectory pooling involves the traversal of a trajectory kernel over the trajectory matrix. Instead of using the summation operation in trajectory convolution, trajectory pooling employs the maximization operation.

During trajectory pooling, when the kernel intersects with elements in the trajectory matrix, it will select the max element from the set of hovered elements. This pooling

approach allows the extraction of key features while mitigating the impact of data sampling rate imbalances.

Given a trajectory matrix $M'_{p \times q}$ after convolution, and a trajectory kernel $\mathcal{K}_{k \times k}$, the new trajectory matrix $\hat{M}_{(p-k+1) \times (q-k+1)}$ after the pooling is

$$\hat{M}_{i,j} = \begin{cases} \frac{1}{1+e^{-x_{i,j}}}, \text{if } x_{i,j} > 0. \\ 0, \text{ otherwise.} \end{cases} \tag{3}$$

where $x_{i,j}$ is the maximum element from $\{M'_{u,v} | i \leq u \leq i+k-1, j \leq q \leq j+k-1\}$. In this representation, each element in the trajectory matrix signifies the likelihood of an object being present within a latent region.

### 4.3. Similarity Computation

Considering two trajectories $Tra_i$ and $Tra_j$ in the same spatial space, they are represented as two matrices $\hat{M}^i_{p \times q}$ and $\hat{M}^j_{p \times q}$ respectively after the trajectory convolution and pooling operations. Each element in the trajectory matrix indicates the possibility that an object is located in a latent region, and we can compute the similarity of two trajectories based on the co-occurrence in the latent regions. To quantify the distance between the two trajectories, we utilize the expression $\sum_{u=1}^{p} \sum_{v=1}^{q} (\hat{M}^i_{u,v} - \hat{M}^j_{u,v})$. This calculation captures the distance between the corresponding elements of the trajectory matrices. However, it is important to note that the lengths of the trajectories may differ. A longer trajectory would yield more locations, resulting in a trajectory matrix with a greater number of non-zero elements.

To address this discrepancy in trajectory lengths, we propose employing the average distance to reflect the similarity between the two trajectories:

$$Dis(Tra_i, Tra_j) = \frac{\sum_{u=1}^{p} \sum_{v=1}^{q} (\hat{M}^i_{u,v} - \hat{M}^j_{u,v})}{S}, \tag{4}$$

where $S = |\{(u,v) | \hat{M}^i_{u,v} \neq 0 \text{ or } \hat{M}^i_{u,v} \neq 0\}|$ is the size of the union of non-zero elements in the two trajectory matrices. By taking the average, it ensures that the similarity assessment is not biased toward trajectories of different lengths. In our work, the distance between two trajectories ranges from 0 to 1. Two trajectories are more similar if the distance between them is closer to 0 and less similar if the distance is closer to 1.

### 4.4. Trajectory Index

During trajectory convolution and pooling, we observe that an element in the trajectory matrix only affects the result of the new trajectory matrix when it intersects with the trajectory kernel. Given a trajectory kernel $\mathcal{K}_{k \times k}$, an element in a trajectory matrix will only affect at most $k \times k$ elements in the new trajectory matrix. Thus, it is not necessary to apply the convolution operation at each location or to compare the difference of elements at each location. For example, the convolution operations in Figure 2a,d are unnecessary as the elements interacting with the trajectory kernel are all 0 and, thus, have no effect on the new trajectory matrix. To improve the computational efficiency of convolution and similarity measurement, we devise a trajectory index based on these characteristics.

Given a trajectory matrix $M_{p \times q}$ and a trajectory kernel $\mathcal{K}_{k \times k}$, for each element $M_{i,j} > 0$, it will affect the value of $M'_{i-k+a,j-k+b}$, where $M'$ is the new trajectory matrix initialized with a zero matrix, $1 \leq a \leq 2k-1$ and $1 \leq b \leq 2k-1$, $1 \leq i-k+a \leq p-k+1$ and $1 \leq j-k+b \leq q-k+1$. Thus, we can continue indexing the elements that are non-zero in a trajectory matrix, and only update the value of elements in a new trajectory matrix, which are affected. During trajectory convolution, for an element $M_{i,j} > 0$, $M_{i,j}$ will be added to $M'_{i-k+a,j-k+b}$ in the new trajectory matrix. In the trajectory pooling layer, when visiting a element $M'_{i,j} > 0$, if $M'_{i,j} > \hat{M}_{i-k+a,j-k+b}$, $\hat{M}_{i-k+a,j-k+b}$ will be replaced by $M'_{i,j}$, where $\hat{M}$ is

a zero matrix with size $(p - 2k - 2) \times (q - 2k - 2)$. After visiting all the non-zero elements in $\hat{M}$, we transform the non-zero element into

$$\hat{M}_{u,v} = \frac{1}{1 + e^{-\hat{M}_{u,v}}}. \tag{5}$$

When calculating the distance between two trajectories, we only consider the differences between the non-zero elements in the respective trajectory matrices. Since the non-zero elements have been indexed, it is not necessary to traverse all elements in the trajectory matrix. This indexing strategy significantly improves the efficiency of both feature extraction operations and similarity comparison.

Given two trajectories $T_a$ and $T_b$, assume that their trajectory matrices are $M^a$ and $M^b$, respectively. There are $O(k^2(|T_a| + |T_b|))$ summation or maximization operations for a trajectory matrix, where $k$ is the size of the trajectory kernel in the Contra. Similarly, there are $O(k^2(|T_a| + |T_b|))$ difference operations when computing the distance between two trajectory matrices. Since $k$ is set to be a small constant for the Contra ($k = 3$ in this work), the time complexity for trajectory convolution and similarity computing is $O(|T_a| + |T_b|)$. Notably, the feature extraction can be performed offline to further improve efficiency. Consequently, the similarity between two trajectories can be compared in linear time.

## 5. Illustrative Experimental Results

In this section, we evaluate the effectiveness, efficiency, and scalability of our proposed approach on two real trajectory datasets. We introduce the datasets and the baseline approaches used for experiments in Section 5.1. To evaluate the effectiveness of the approaches, we conduct the experiments from two aspects—self-similarity and cross-similarity—which were widely adopted in the previous works [19,22,29]. Performance results for self-similarity and cross-similarity are are shown in Sections 5.2 and 5.3. Moreover, we present the evaluation results for the efficiency and scalability of our approach in Section 5.4. The effects of parameters for the proposed approach are discussed in Section 5.5.

### 5.1. Datasets and Baselines

We conducted our experiments on two real-world taxi datasets. The statistics of the datasets are shown in Table 1. The first dataset (Porto dataset) (https://www.kaggle.com/datasets/crailtap/taxi-trajectory accessed on 17 October 2023) contains 1,233,766 trajectories, collected by 422 taxis running in the city of Porto, Portugal, over 12 months. Each taxi reports its location every 15 seconds. The average duration of these trajectories is 783.45 s.

The second dataset (the Beijing dataset) [37,38] consists of GPS trajectories from 10,257 taxis over a one-week period in Beijing. The average sampling interval for this dataset is 174.36 s. We partition a trajectory into two if the time gap between two consecutive points is more than 15 min, leading to 333,948 trajectories. The average duration of these trajectories is 8473.26 seconds. To ensure meaningful analysis, trajectories with a length of less than 30 data points were excluded, allowing us to sample sub-trajectories at different rates and evaluate the effect of low and heterogeneous data sampling rates. Following the experimental settings in prior works [19,22], we conducted experiments using a subset of 100,000 trajectories in our work.

**Table 1.** Dataset statics.

| | Number of Trajectory | Average Time Interval (s) | Average Duration (s) |
|---|---|---|---|
| Porto dataset | 1,233,766 | 15.00 | 783.45 |
| Beijing dataset | 333,948 | 174.36 | 8473.26 |

We compare our proposed approach, Contra, with three existing works, namely EDwP [19], CATS [25] and LCSS [20]. LCSS is a widely adopted approach for measuring sequence similarity and has been extensively used in trajectory similarity analysis. EDwP and CATS are state-of-the-art methods specifically designed to address challenges posed by heterogeneous and low sampling rates in trajectory similarity analysis. We excluded EDR [21] from our comparison since EDwP is an extension of EDR and has shown superior performance compared to the traditional EDR approach. Similarly, we did not include DTW [18] as it has been demonstrated to have similar performance to EDR in trajectory similarity measurements [19].

For the evaluation, we set the parameter $\epsilon$ in LCSS and EDR to 0.0025 based on our experimental results. The spatial threshold in CATS was set to 0.002, as it achieved the best performance in our experiments. In the case of Contra, we configured the trajectory kernel size as 3, and the default grid size as 100 m. The effect of grid sizes will be discussed in subsequent experiments. Contra, CATS, and LCSS were implemented in Python, while the authors of EDwP provided the implementation in Java, which can be obtained from their website (http:www.cse.iitd.ac.in/~sayan/software.html accessed on 17 Ocotber 2023).

### 5.2. Effectiveness of Self-Similarity

An effective trajectory similarity measure should identify trajectories sampled from the same path. Thus, we evaluate the performance of Contra and other baseline approaches using self-similarity, which is widely discussed in prior works [22,32].

Given two sets of trajectories $D^1$ and $D^2$, consider any trajectory pair $Tra_i^1 \in D^1$ and $Traj_i^2 \in D^2$. $Tra_i^1$ and $Tra_i^2$ are sampled from the same path. We measure the similarity of $Tra_i^1$ and any trajectories in $D^2$. Then we sort the trajectories in $D^2$ with respect to the similarity in descending order, and use $r_i$ to denote the rank of $Tra_i^2$. If $r_i = 1$, the precision for $Tra_i^1$ is defined as $p_i = 1$, and $p_i = 0$ otherwise. Then, the precision $P$ for all testing trajectories is defined as

$$P = \frac{\sum_{i=1}^{n} p_i}{n}. \tag{6}$$

Furthermore, the mean rank is defined as the average of all $r_i$:

$$MR = \frac{\sum_{i=1}^{n} r_i}{n}. \tag{7}$$

Both metrics could reveal the effectiveness of similarity measures in finding similar trajectories, assessing the measures from different aspects. The precision metric cares about whether a measure could identify the exact trajectory sampled from the same path for a trajectory, while the mean rank metric considers the ranking among the trajectory dataset. The higher precision and lower mean rank indicate the better effectiveness of the measure. The mean rank will be 1 when the precision is 100%.

**Data construction.** To overcome the lack of ground truth, we constructed the evaluation dataset following prior works [22,32]. As shown in Figure 3, for each trajectory $Tra_i$ in a dataset, we generated two sub-trajectories, $Tra_i^1$ and $Tra_i^2$, by alternately taking points from it, and used them to construct two new datasets, $D^1 = \{Tra_i^1 | i = 1, 2, \cdots, n\}$ and $D^2 = \{Tra_i^2 | i = 1, 2, \cdots, n\}$. In the two new datasets, $Tra_i^1 \in D^1$ and $Tra_i^2 \in D^2$ are sampled from the same trajectory $Tra_i$. We first randomly selected 6000 trajectories from the Porto dataset and the Beijing dataset. Next, we constructed the evaluation datasets based on the selected Porto dataset $D_P$ and the Beijing dataset $D_B$, and obtained two pairs of new datasets, $(D_P^1, D_P^2)$ and $(D_B^1, D_B^2)$.

**Effects of different data sampling rates.** We assessed the performance of various methods under different sampling rates. To achieve this, we applied downsampling to the trajectories in both $D^1$ and $D^2$ using dropping rates of $[0, 0.1, 0.2, 0.3, 0.4, 0.5, 0.6]$, resulting in trajectories with varying sampling rates. As a result, we obtained a total of 14 datasets, comprising 7 pairs, where each pair consists of two datasets downsampled with the same data-dropping rate. Next, we conducted a comparison of trajectory similarity within

each pair. For every trajectory in the dataset downsampled from $D^1$, we calculated its similarity with all trajectories from the dataset downsampled from $D^2$ at the corresponding data-dropping rate.

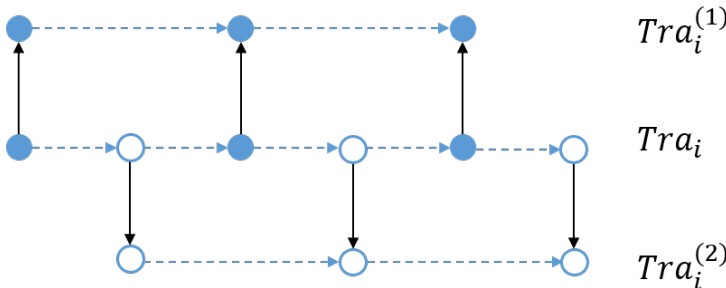

**Figure 3.** Sample two sub-trajectories from a trajectory for ground truth construction.

The precision and mean rank on the Porto dataset for different data sampling rates are depicted in Figures 4a and 5a, respectively. The experimental results clearly demonstrate that our proposed approach consistently outperforms all other methods in terms of precision and mean rank. For trajectories without downsampling (dropping rate = 0), Contra achieves an impressive precision of up to 96% and a mean rank of approximately 1.06, showing significant improvements compared to the state-of-the-art methods, EDwP and CATS. EDwP and CATS exhibit precision values of 85% and 61%, respectively, with mean ranks of 1.67 and 5.18. Notably, Contra demonstrates an 11% improvement in precision over EDwP and a remarkable 35% improvement over CATS. Furthermore, Contra surpasses EDwP by 50% and CATS by nearly five times in terms of the mean rank. In contrast, LCSS performs significantly worse, with precision values of approximately 5% and mean ranks of around 30.

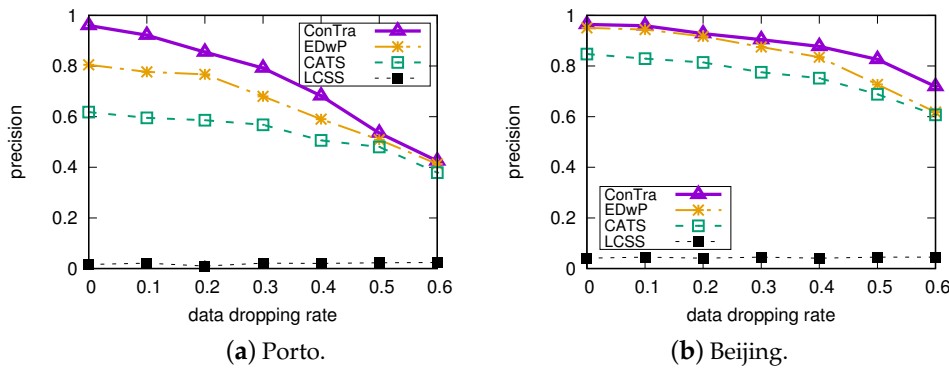

(**a**) Porto.  (**b**) Beijing.

**Figure 4.** Precision versus low data sampling rates.

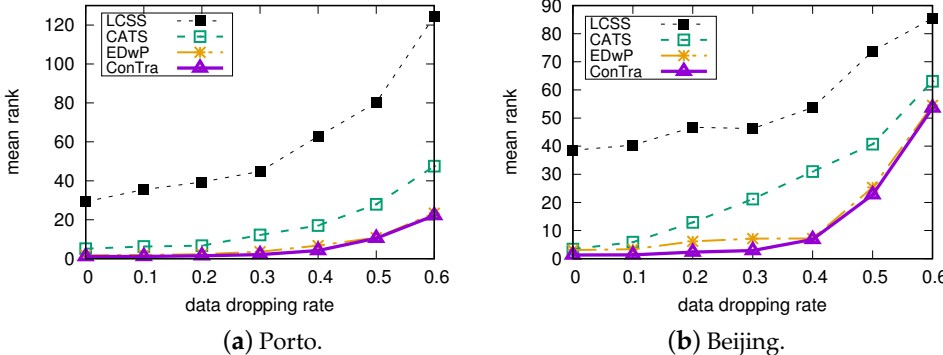

(**a**) Porto.  (**b**) Beijing.

**Figure 5.** Mean rank versus low data sampling rates.

In this experiment, a higher dropping rate indicates a lower sampling rate. As shown in Figures 4a and 5a, with the increase in the dropping rate, the performance of all methods decline since the data sampling rates become lower. However, even at low sampling rates, our approach maintains higher precision and lower mean rank compared to any other methods. The experimental results for the Beijing dataset, depicted in Figures 4b and 5b, are consistent with those of the Porto dataset. Contra also achieves the best performance in both metrics for the Beijing dataset, indicating the robustness of our approach across different datasets and varying data sampling rates.

**Effect of heterogeneous data sampling rates**. In real-world applications, data sampling rates across different devices often vary, leading to heterogeneity. Therefore, we investigate the effect of heterogeneous data sampling rates on the performance of different methods. To accomplish this, we apply downsampling to the trajectories in $D_2$ using data-dropping rates of $[0, 0.1, 0.2, 0.3, 0.4, 0.5, 0.6]$, while trajectories in $D_1$ remain without downsampling. As a result, we obtained six new datasets, each downsampled from $D_2$ and featuring a heterogeneous sampling rate compared to $D_1$. Subsequently, we calculate the similarity between each trajectory in $D_1$ and any trajectories from the datasets downsampled from $D_2$ at different data-dropping rates. We use precision and mean rank as evaluation metrics.

The precision and mean rank versus heterogeneous data sampling rates on the Porto dataset are presented in Figures 6a and 7a, respectively. In both figures, the data-dropping rate indicates the difference in the data sampling rates of the two compared trajectories. The larger the data-dropping rate, the larger the difference in the data sampling rates of two trajectories. The figures demonstrate that Contra consistently outperforms all other baseline approaches across all data sampling rates, highlighting its superior performance in addressing the issue of heterogeneous data sampling rates. In Figure 6a, the precision of Contra is above 75%, even with a data-dropping rate of 0.3. Notably, state-of-the-art methods, such as EDwP and CATS, perform better than the traditional approach LCSS, which is consistent with the results in previous works. As the data-dropping rate increases, indicating a larger disparity in sampling rates between trajectories, the performance of all approaches declines. This decline occurs because a larger difference in sampling rates makes it more challenging to measure the similarity between trajectories. Similar trends can be observed in the mean rank results, as depicted in Figure 7a. Contra consistently outperforms other approaches in terms of mean rank, while EDwP shows comparable performance when the data sampling rate of trajectories in dataset $D^2$ is low.

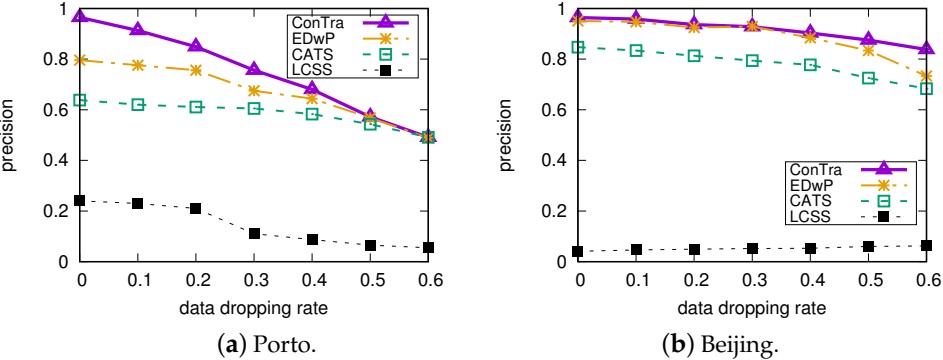

**(a)** Porto.          **(b)** Beijing.

**Figure 6.** Precision versus heterogeneous data sampling rates.

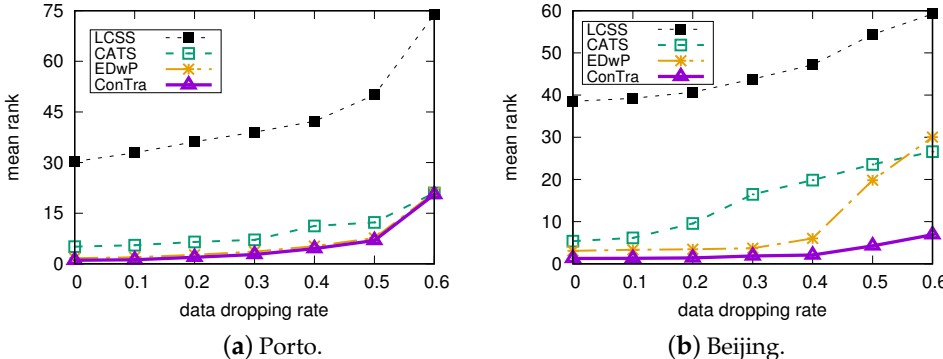

**Figure 7.** Mean rank versus heterogeneous data sampling rates.

The results of precision and mean rank over the Beijing dataset are shown in Figures 6b and 7b. Contra significantly outperforms other approaches on the Beijing dataset. In Figure 6b, Contra maintains precision values above 80% even at a high data-dropping rate (e.g., 0.6), whereas both EDwP and CATS fall below 70%. Similarly, in Figure 7b, Contra achieves a mean rank below 10 at a data-dropping rate of 0.6, outperforming EDwP and CATS by a factor of three and LCSS by a factor of six. The consistent results across both datasets demonstrate the robustness and general applicability of Contra for trajectories with heterogeneous data sampling rates.

**Effect of location noise**. A reliable similarity measure should be able to accurately identify similar trajectories even in the presence of location noise. To evaluate the performance of methods under different degrees of location noise, we distort the location in a trajectory by adding Gaussian noise with a different location noise $w(m)$, as follows,

$$
\begin{aligned}
x_i &= x_i + w \cdot d_x, d_x \sim Gaussian(0,1), \\
y_i &= y_i + w \cdot d_y, d_y \sim Gaussian(0,1).
\end{aligned}
\tag{8}
$$

We distort each trajectory in datasets $D^1$ and $D^2$ using location noise values of [0 m, 5 m, 10 m, 15 m, 20 m, 25 m, 30 m]. This process results in two new sets of distorted datasets, namely $D_i^1$ and $D_i^2$, for each location noise value $w_i$. Consequently, we obtain a total of 14 datasets comprising 7 pairs, where trajectories in each pair are distorted by the same location noise level. Next, we proceed to compare the similarity between each trajectory in $D_i^1$ and all trajectories within $D_i^2$.

The precision and mean rank results for the Porto dataset are presented in Figure 8a and Figure 9a, respectively. Among all the approaches, Contra achieves the highest performance on both metrics. As shown in Figure 8a, with even the most severe noise of 30 m, the precision is up to 92.7%. Notably, Contra achieves approximately a 20% improvement in precision compared to EDwP and a remarkable 40% improvement compared to CATS. Conversely, the precision of the LCSS approach performs poorly in the experiment. Figure 9a illustrates the mean rank results, where Contra consistently achieves a mean rank close to 1 for all levels of location noise. Even in the worst-case scenario with a location noise of 30 m, Contra maintains a mean rank of less than 1.13. In terms of mean rank, Contra consistently outperforms EDwP by 20% and CATS by approximately 30 times. As the severity of the location noise increases, the decline in performance on both metrics is not significant, highlighting the robustness of Contra in the presence of location noise.

The experiment results over the Beijing dataset are presented in Figures 8b and 9b, demonstrating the superior performance of Contra on both metrics. These results clearly indicate that our proposed approach, Contra, is more effective and robust in handling location noise compared to existing approaches.

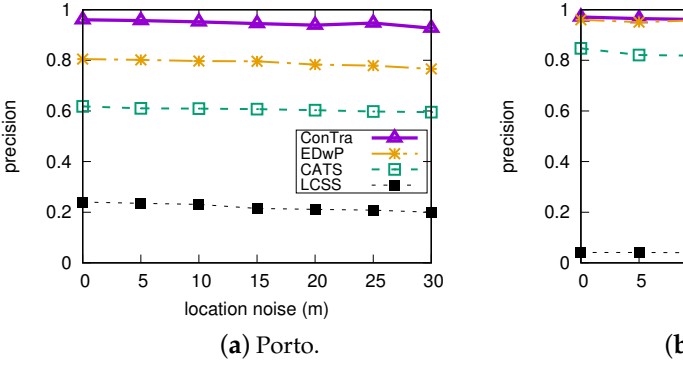

**Figure 8.** Precision versus location noise.

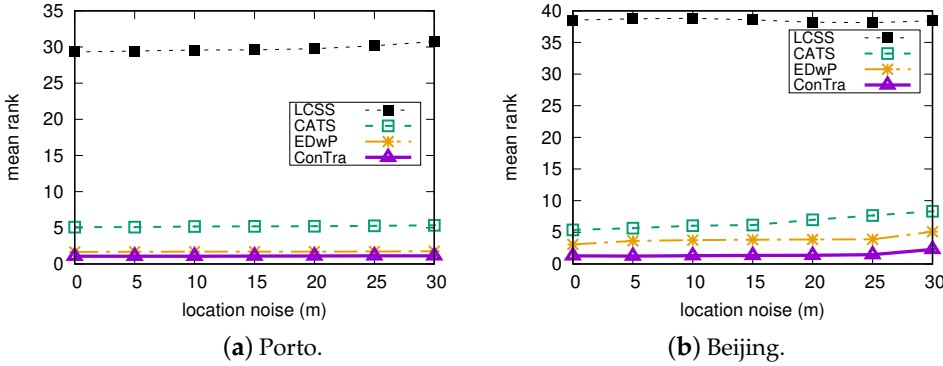

**Figure 9.** Mean rank versus location noise.

### 5.3. Effectiveness of Cross-Similarity

An effective trajectory measure should preserve the distance between two distinct trajectories (cross-similarity), regardless of variations in data sampling rate and location noise [22,29]. In this experiment, we randomly select 10,000 pairs of trajectories from the dataset. For each trajectory pair, denoted as $Tra_i$ and $Tra_j$, we calculate the ground truth distance $Dis(Tra_i, Tra_j)$ between them. Subsequently, we calculate the distance between $Tra_i$ and a modified version of $Tra_j$, referred to as $Tra_j'$. The modification involves either removing locations from $Tra_j$ with a data-dropping rate denoted as $\theta_i$ or adding location noise denoted as $w_i$. We use the cross-similarity deviation as the metric, which is defined in prior work [29] as follows,

$$CD = \frac{|dis(Tra_i, Tra_j') - dis(Tra_i, Tra_j)|}{dis(Tra_i, Tra_j)}. \tag{9}$$

A smaller cross-similarity deviation indicates closer proximity to the ground truth distance, highlighting better preservation of cross-similarity.

**Effect of the data sampling rate.** To evaluate the cross-similarity for heterogeneous data sampling rates, the data-dropping rate $\theta_i$ is set to be $[0.1, 0.2, 0.3, 0.4, 0.5, 0.6]$ for the experiment. The corresponding results for the Porto dataset and the Beijing dataset are presented in Figure 10a and Figure 10b, respectively. The cross-similarity deviation of Contra is much smaller than that of CATS and LCSS on both datasets. The difference between Contra and EDwP is not significant on the Porto dataset, while Contra outperforms EDwP slightly on the Beijing dataset. As the data-dropping rate increases, the cross-similarity deviation of CATS and LCSS increases dramatically, while that of Contra and EDwP increases slowly. The consistency of these results across both datasets reinforces

the effectiveness of Contra in accurately preserving distances even when confronted with varying data sampling rates.

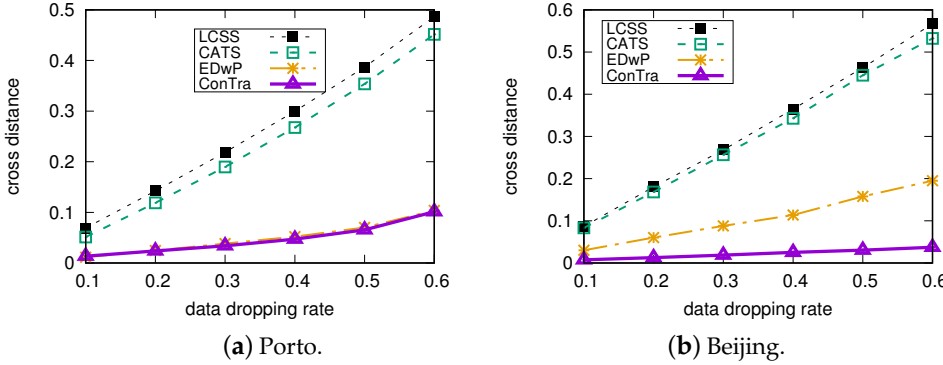

**Figure 10.** Cross-similarity deviation versus different data sampling rates.

**Effect of location noise**. The location noise $w_i$ is set to be [5 m, 10 m, 15 m, 20 m, 25 m, 30 m] to investigate the effect of location noise on cross-similarity. Figure 11a,b show the results on the Porto dataset and the Beijing dataset. On both datasets, as the severity of location noise increases, there is a rapid escalation in the cross-similarity deviation for CATS and LCSS. Conversely, the increment in cross-similarity deviation for Contra and EDwP is not as significant. The deviation of both Contra and EDwP is much smaller than that of CATS and LCSS, which reveals the remarkable ability of Contra and EDwP to effectively preserve distances even when confronted with varying levels of location noise.

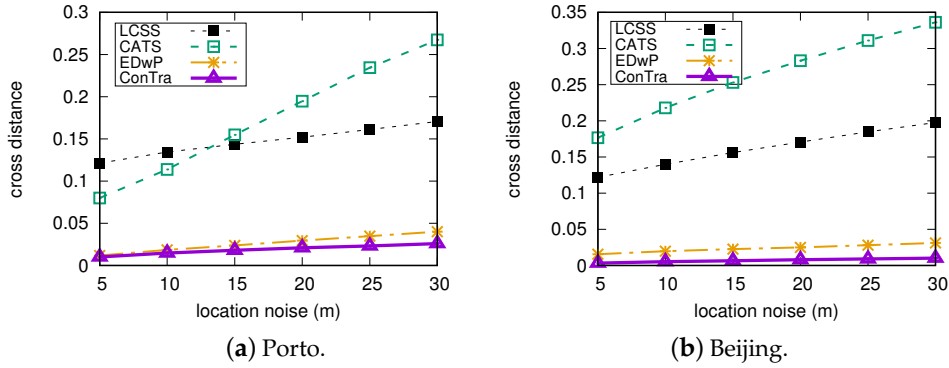

**Figure 11.** Cross-similarity deviation versus different location noise.

### 5.4. Efficiency and Scalability

To evaluate the efficiency and scalability of Contra, we conducted a comparative analysis between Contra and other baseline approaches. The evaluation focused on the running time required to calculate the similarity of a varying number of trajectory pairs, specifically $n = \{20,000, 40,000, 60,000, 80,000, 100,000\}$. The results obtained from the Porto dataset and Beijing dataset are presented in Figure 12a and Figure 12b, respectively.

As observed in Figure 12a, the running time on the Porto dataset increases proportionally with the growth in data size. Notably, Contra consistently outperforms all other baseline approaches in terms of computational efficiency across the different data sizes. Moreover, the performance gap between Contra and the baselines becomes more significant as the data size increases. The trends are similarly reflected in the results obtained from the Beijing dataset (Figure 12b). These results show the efficiency and robustness of our proposed approach, as demonstrated on both the Porto and Beijing datasets.

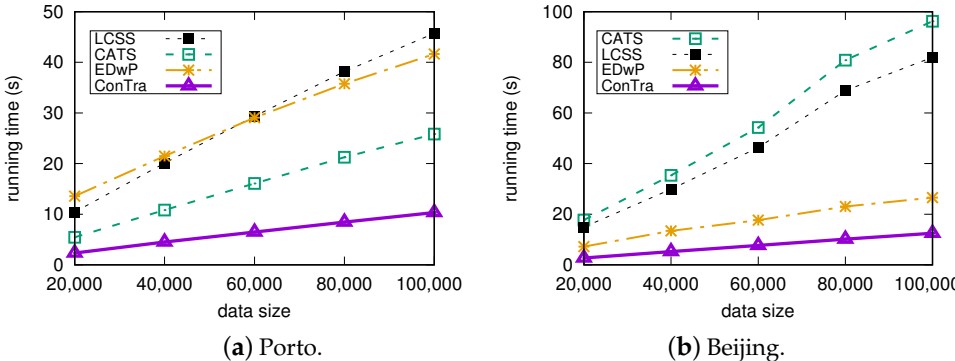

(**a**) Porto.　　　　　　　　　　　　(**b**) Beijing.

**Figure 12.** Running time versus different data sizes.

### 5.5. Effects of Different Parameters

We evaluate the effects of grid size, trajectory kernel size, and data size on the performance of our proposed approach in our experiments.

**Effect of grid size.** Contra uses sequences of grids to represent trajectories. Intuitively, using large grid sizes enhances the tolerance to noise and low sampling rates for measuring trajectory similarity. However, employing excessively large grids may lead to the misclassification of dissimilar trajectories as similar, as a majority of their locations might be mapped to the same grids. To evaluate the performance of Contra under different grid sizes, we conducted experiments using grid sizes of [50 m, 60 m, 70 m, 80 m, 90 m, 100 m, 110 m, 120 m] and evaluated Contra using the metrics of precision and mean rank. Figure 13a illustrates the precision results for both datasets as the grid size varies. It can be observed that precision initially increases with larger grid sizes, reaching a peak point. Subsequently, precision begins to decline. For the Porto dataset, Contra achieves the best performance when the grid size is set to 80 m, and for the Beijing dataset, the optimal grid size for Contra is 100 m. Notably, Contra maintains consistently high precision across different grid sizes, indicating its robustness in handling variations in grid size. Similar trends can be observed in the mean rank metric, as depicted in Figure 13b. These results further reinforce the findings from the precision metric, highlighting the effectiveness of Contra across different grid sizes.

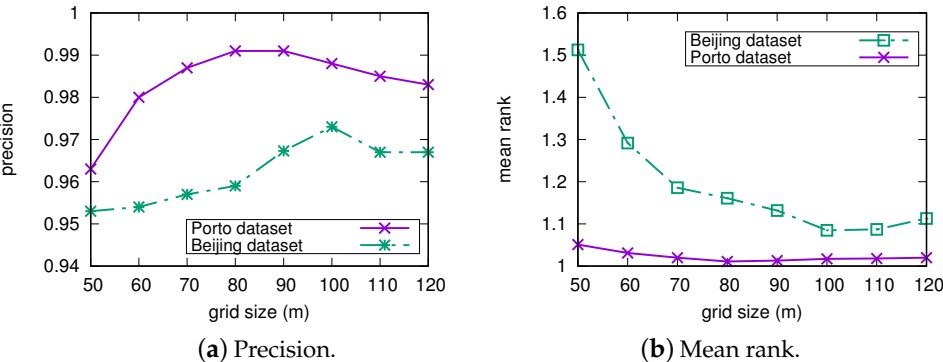

(**a**) Precision.　　　　　　　　　　　(**b**) Mean rank.

**Figure 13.** Performance versus different grid sizes.

**Effect of kernel size.** During the trajectory convolution and pooling operations, Contra employs a trajectory kernel to extract high-level features. To investigate the effects of different kernel sizes on Contra's performance, we conducted experiments using a range of kernel sizes, including [2, 3, 4, 5, 6, 7, 8]. Precision and mean rank are used as evaluation metrics. The experimental findings are presented in Figure 14a,b. The performance of Contra exhibits a trend of initially increasing and then declining as the kernel size varies. Figure 14a illustrates the precision results, indicating that Contra achieves the highest

precision when the kernel size is set to 3 for the Porto dataset and 5 for the Beijing dataset. Remarkably, Contra maintains consistently high precision even when the filter size changes. Similar trends can be observed in the mean rank metric, as depicted in Figure 14b.

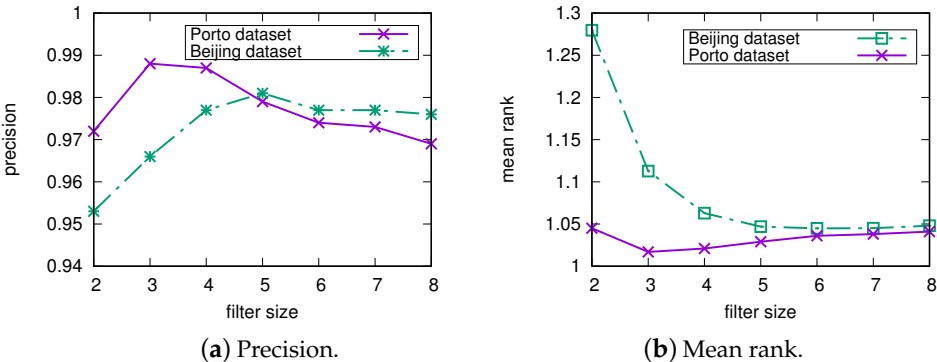

(**a**) Precision.  (**b**) Mean rank.

**Figure 14.** Performance versus different kernel sizes.

**Effect of data size** In order to evaluate the robustness of Contra's performance with increasing data size, we conducted experiments to evaluate its effectiveness in handling varying dataset sizes. The data size was systematically set to [2000, 4000, 6000, 8000, 10,000] for our evaluation, with precision and mean rank serving as the evaluation metrics.

The results, presented in Figure 15a,b, reveal important insights. As depicted in Figure 15a, the precision of Contra exhibits a decline as the data size increases. Nevertheless, it is noteworthy that Contra consistently maintains a high level of precision, even with larger datasets. Additionally, the rate of decline in precision becomes slower as the data size increases. This indicates that the performance degradation is less pronounced as the dataset grows in size. Similarly, Figure 15b demonstrates a comparable trend in the mean rank metric. The growth in mean rank decelerates with the increment of data size, and Contra consistently achieves a low mean rank across different data sizes.

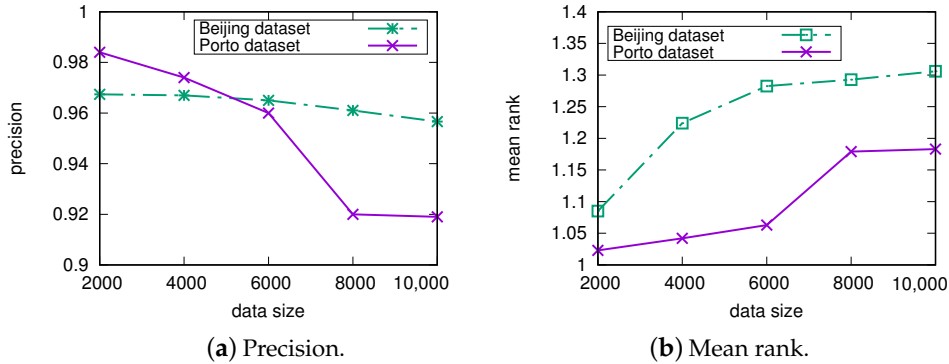

(**a**) Precision.  (**b**) Mean rank.

**Figure 15.** Performance versus different data sizes.

## 6. Conclusions

This paper introduces Contra, a novel and highly effective similarity measure specifically designed to address the challenges associated with uncertain trajectories. Contra measures trajectory similarity by quantifying the spatial overlap between trajectories, where two trajectories are considered similar if their shapes are similar. By extracting high-level spatial features from the trajectories, Contra enables accurate and efficient comparisons, effectively handling issues of location noise, low data sampling rates, and heterogeneous data sampling rates. Furthermore, the incorporation of an efficient trajectory index in Contra significantly enhances the efficiency of both feature extraction and similarity comparison processes. The extensive experiments conducted on two taxi datasets have demonstrated the effectiveness, efficiency, and robustness of our proposed Contra.

**Author Contributions:** Conceptualization, Guanyao Li, Xingdong Deng, Jianmin Xu and Yang Liu; methodology, Guanyao Li, Xingdong Deng, Jianmin Xu, Yang Liu and Ji Zhang; investigation, Guanyao Li, Ji Zhang, Simin Xiong and Feng Gao; writing—original draft preparation, Guanyao Li, Ji Zhang, Simin Xiong and Feng Gao; writing—review and editing, Guanyao Li, Xingdong Deng, Jianmin Xu and Yang Liu; supervision, Xingdong Deng, Jianmin Xu and Yang Liu; project administration, Xingdong Deng and Yang Liu; funding acquisition, Xingdong Deng and Yang Liu. All authors have read and agreed to the published version of the manuscript.

**Funding:** This research was funded by the Guangdong Enterprise Key Laboratory for Urban Sensing, Monitoring, and Early Warning (no. 2020B121202019).

**Data Availability Statement:** Not applicable.

**Conflicts of Interest:** The authors declare no conflict of interest.

**Abbreviations**

| | |
|---|---|
| MR | mean rank |
| CNN | convolution neural network |
| Contra | convolution-based similarity measure for uncertain trajectories |

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
