# Peer review of "An Efficient and Accurate Convolution-Based Similarity Measure for Uncertain Trajectories"

_ijgi, doi:10.3390/ijgi12100432_

Round 1

Reviewer 1 Report

Please refer to the attached review report.

Author Response

We thank the reviewer for the constructive and encouraging comments.  Please see the attachment.

Reviewer 2 Report

An efficient and accurate convolution-based similarity measure, termed Contra, is presented for uncertain trajectories. It uses a series of convolution and pooling operations to extract the trajectory features and compare the similarities among them. A trajectory index strategy has also been proposed to enhance computational efficiency. Some comprehensive experiments have been conducted to verify the performance of Contra. However, there exist some problems, as follows: 

  1. 1. The system overview in Figure 1 is too simple and confusing; specific components improved by different mechanisms presented in this study should be enhanced, such as location mapping, trajectory feature extraction using convolution and pooling, trajectory index generation, and similarity comparison. 

  1. 2. Based on the challenges of heterogeneous sampling rates, low sampling rates, and location noise, this study should enhance the contributions and refine them corresponding to these challenges. Moreover, the extensive experiments on real datasets cannot be considered a contribution. 

  1. 3. The related work cannot be summarized well. 

  1. 4. In Section 4, the steps of convolution and pooling should emphasize the characteristics of trajectory. Figure 2 is a general process of convolution operation. Moreover, compared to the general methods, the authors should enhance the improvements. 

  1. 5. Figure 3 (b) should be corrected as precision comparison of the Beijing dataset. There are similar errors for “Figure”, for example, “Figure 3(b) and 4(b)” should be corrected as “Figures 3(b) and 4(b)”. Please check the whole manuscript.

The quality of English Language should be enhanced.

Author Response

(The authors gave the same response as above.)

Reviewer 3 Report

This work proposed a novel convlultion-based appraoch for measuring similariry of trajectry. The novelty and well-structured of the paper are appreciated. I have the following suggestions for the authors’ consideration.

1.       The key definition of “uncertain trajectory” should be clarified. I do not think this is well-acknowledged term in relevant field, so the author better give a defnite explanation on this concept. I noticed there are existing concepts looking simiar, such as trajectory uncertainties, movement uncertainties, in “A Bi-LSTM approach for modelling movement uncertainty of crowdsourced human trajectories under complex urban environments”, “A deep-learning approach for modelling pedestrian movement uncertainty in large-scale indoor areas”. Is the “uncertain trajectory” in your work same as the relevant concepts in these works? What exactly are you talking about to differentiate yours from others? This should be clarified. (Also, the above relevant works are missed in the reference list)

2.       It would be better if the rationale of the Contra can be given. What is the reason for the design of Contra? You mentioned some challenge/limitaions of previous works, so how did the Contra address these challenge? Or how did these challenges lead to the specifications of your Contra design? Such explanations should be given to justify your model design.

3.       “Linear processing time” and “Experiments on real datasts” seem not signifnicant contributions for me. Also, the experiments supporting linear processing time should be given and discussed in the paper to support the argument.

4.       I suggest a table given to show the stasticis of the dataset. Also, I think it is more crucial to show the number of trajectories used for the experiments, than the number of taxi.

Author Response

(The authors gave the same response as above.)

Reviewer 4 Report

The authors proposed Contra, a similarity search technique on trajectory data that uses grid-based mapping and convolutional pool. The technique focuses on eliminating the trajectory noise and heterogeneous sampling rate of each recorded points. The authors has presented the methodology and exhaustive experiments and concluded that Contra is efficient for the uncertain trajectories.

However, I believe the manuscript can be improved by addressing some discussion points below.

  1. The development of the deep learning/machine learning techniques towards trajectory similarity could be improved by addressing more or newer works. Please check these references if they are relevant:
    • Trajgat: A graph-based long-term dependency modeling approach for trajectory similarity computation (https://dl.acm.org/doi/abs/10.1145/3534678.3539358)
    • CSTRM: Contrastive Self-Supervised Trajectory Representation Model for trajectory similarity computation (https://www.sciencedirect.com/science/article/abs/pii/S0140366422000019) 
    • X-FIST: Extended flood index for efficient similarity search in massive trajectory dataset (https://www.sciencedirect.com/science/article/pii/S0020025522004947#b0165)
    • Efficient and effective similar subtrajectory search with deep reinforcement learning, (http://www.vldb.org/pvldb/vol13/p2312-wang.pdf)
  2. How to have a good rule of thumb to define s? Experimental result shows a good evaluation, a consideration on the how sparse the recorded trajectories are in the trajectory matrix or how heterogeneous the sampling rate with sample data may give a good approximation before experimenting on the real dataset. The authors also might be able to have a good grid sizing definition based on the context of trajectory: indoor/outdoor, semantic information of the mobility, etc.
  3. How do we maintain the order of points in the trajectory matrix?
  4. LCSS and EDwP might not be good baseline towards Contra as they consider the order of the trajectory as the distance measurement. Thus, the comparison might not be relevant to Contra. The author might propose a modified version of LCSS and EDwP to gain fairer comparison towards Contra.

The English presentation is adequate to be presented as it is.

Author Response

(The authors gave the same response as above.)

Round 2

Reviewer 2 Report

All comments have been addressed.

Written English should be checked carefully. 

Author Response

We thank the reviewer for the positive comments.

Reviewer 3 Report

The authors have addressed the concerns

Author Response

(The authors gave the same response as above.)

Reviewer 4 Report

I am satisfied with the current state of the manuscript.
Just some notes that I would like to suggest:

- the authors can remove the suggested references before if they feel the references are irrelevant.

- emphasize the focus of Contra study is to identify the similarity of the shape of trajectories and ignoring the time/order relevancy of each record in the trajectory.

-

Author Response

We thank the reviewer for the positive comments. Please see the attachment.
